# The Role of the Epidermal Growth Factor Receptor in Diabetic Kidney Disease

**DOI:** 10.3390/cells11213416

**Published:** 2022-10-28

**Authors:** Raymond C. Harris

**Affiliations:** 1Division of Nephrology and Hypertension, Department of Medicine, Vanderbilt Center for Kidney Disease, Vanderbilt University Medical Center, Nashville, TN 37232, USA; ray.harris@vumc.org; Tel.: +1-615-202-9426; 2Tennessee and Veterans Affairs, Nashville, TN 37232, USA

**Keywords:** EGF, kidney, diabetes, nephropathy

## Abstract

The epidermal growth factor receptor (EGFR) is expressed in numerous cell types in the adult mammalian kidney and is activated by a family of EGF-like ligands. *EGFR* activation has been implicated in a variety of physiologic and pathophysiologic functions. There is increasing evidence that aberrant *EGFR* activation is a mediator of progressive kidney injury in diabetic kidney disease. This review will highlight recent studies indicating its potential role and mechanisms of injury of both glomerular and tubular cells in development and progression of diabetic kidney disease.

## 1. The Epidermal Growth Factor Receptor and Its Ligands

The epidermal growth factor receptor (EGFR; aka ErbB1 or HER1) is a member of the ErbB/HER family of receptor tyrosine kinases, which also includes ErbB2 (HER2), ErbB3 (HER3) and ErbB4 (HER4). All four ErbBs have a common structure, with an extracellular ligand-binding domain, a single membrane-spanning region, a homologic cytoplasmic protein tyrosine kinase domain and a C-terminal tail with multiple phosphorylation sites. *EGFR* is activated by the ligands of the EGF-related peptide growth factor family [1]. These ligands include EGF, amphiregulin, transforming growth factor-alpha (TGF-α) betacellulin, heparin-binding EGF (HB-EGF) epiregulin and epigen [2,3]. In addition, a study has suggested that connecting tissue growth factor (CTGF) can bind to and activate *EGFR* [4].

## 2. Expression of *EGFR* in the Kidney

The *EGFR* is expressed in numerous cell types in the adult mammalian kidney, including podocytes, endothelial cells and mesangial cells in the glomerulus and in multiple tubule segments [3]. It is also expressed in cells in the interstitium, including medullary interstitial cells as well as in resident and infiltrating immune cells. The *EGFR* has been implicated in regulation of both physiologic and pathophysiologic functions in the adult kidney [3,5,6,7,8].

## 3. Mechanisms of Dysfunctional Expression and Activation of *EGFR* in Diabetic Nephropathy

There is upregulation of expression and activation of *EGFR* in experimental models of diabetic kidney injury and in cultured renal cells exposed to high glucose [9,10,11,12,13]. A recent study has also identified a SNP of an enhancer located in the *EGFR* gene associated with upregulation of *EGFR* expression in type II diabetes [14]. In addition, a number of *EGFR* ligands, including TGF-α, HB-EGF and amphiregulin, have been reported to increase in experimental models of diabetic nephropathy [10,13,15].

## 4. Effect of *EGFR* Inhibition to Ameliorate Progression of Diabetic Nephropathy

In a streptozotocin (STZ) model of type I diabetes in mice, wild type mice developed mesangial expansion and moderate albuminuria after 24 weeks of diabetes, and mice with selective deletion of endothelial nitric oxide synthase (eNOS^–/–^) had markedly exacerbated development of diabetic nephropathy [16]. We treated both wild type and eNOS-deficient diabetic mice with the *EGFR* tyrosine kinase inhibitor, erlotinib, for 22 weeks. Erlotinib decreased both albuminuria and mesangial expansion and decreased glomerulosclerosis in the STZ-eNOS^–/–^ mice. Similar studies in diabetic rats reported that a different *EGFR* tyrosine kinase inhibitor, PKI 166 also attenuated early glomerular enlargement, decreased proteinuria and preserved podocyte number [17].

The eNOS^–/–^
*db/db* mouse model is an accelerated model of diabetic nephropathy in an obese type 2 model of diabetes secondary to leptin receptor deficiency. These mice develop significant functional and structural kidney injury within the first 20 weeks of life [18]. Inhibition of *EGFR* activation by erlotinib in the kidney of eNOS^–/–^
*db/db* mice was confirmed by inhibition of *EGFR* phosphorylation and inhibition of activation of ERK1/2 [19]. The eNOS^–/–^
*db/db* mice have increased blood pressure compared to wild type, and erlotinib administration did not decrease blood pressure. However, erlotinib did prevent further increases in albuminuria and resulted in significantly less glomerulosclerosis. In addition, there was relative preservation of podocyte number in response to erlotinib treatment. Furthermore, there was decreased tubulointerstitial injury and fibrosis, indicated by deceased expression of the proximal tubule injury marker, KIM-1, decreased fibrillar extracellular collagen indicated by less Sirius red and collagen I staining and decreased expression of mRNA for components of fibrosis (collagens I&III, fibronectin), decreased myofibroblasts, indicated by less α-SMA immunostaining and decreased expression of profibrotic factors (transforming growth factor beta (TGF-ß), CTGF). There was a marked decrease in renal macrophage infiltration with erlotinib treatment, indicated by decreased F4/80 mRNA expression and decreased F4/80 staining. Erlotinib also decreased T cell infiltration, as indicated by decreased CD3 mRNA and CD8α staining. Erlotinib treatment led to a significant decrease in mRNA for IRF5, a mediator of proinflammatory M1 macrophage phenotype [20] and decreased mRNA for proinflammatory cytokines (iNOS, TNF-α, INF-γ, IL-6).

The Erlotinib treated mice had a slower rate to increase body weight, which was secondary to relatively less increase in the fat tissue mass. Accompanying this decreased weight gain, erlotinib also decreased fasting blood glucose levels and improved glucose tolerance and insulin tolerance tests. The erlotinib-treated mice had preserved pancreatic islet insulin staining, higher fasting blood insulin levels and decreased islet macrophage infiltration. Erlotinib administration also prevented decreases in serum levels of adiponectin, an adipocyte-derived hormone that increases insulin sensitivity.

As a confirmation of the role of *EGFR* activation in mediating diabetic kidney injury, we also studied waved 2 mice, which have a point mutation in *EGFR* that reduces intrinsic tyrosine kinase activity by >90% [21]. Similar to erlotinib inhibition of *EGFR*, homozygous waved 2 mice crossed to eNOS^–/–^db/db mice exhibited decreased gain of body weight, lower fasting blood glucose, preserved pancreatic islet insulin levels, less islet macrophage infiltration, and less glomerulosclerosis. waved 1 mice have markedly decreased expression of the *EGFR* ligand, TGF-α [22]. Similar to erlotinib-treated eNOS^–/–^
*db/db* and waved 2 eNOS^–/–^
*db/db* mice, waved 1 mice crossed to eNOS^–/–^
*db/db* mice also had marked decreases in gain of body weight, fasting blood glucose, islet macrophage infiltration, and glomerulosclerosis and preserved pancreatic insulin levels. Serum and urine TGF-α increases in human diabetic kidney disease, and neutralizing antibodies to TGF-α were found to slow progression in models of accelerated diabetic kidney disease [15].

## 5. The Role of *EGFR* Activation in Podocyte Injury in Diabetic Nephropathy

Glomerular podocytes are highly specialized cells characterized by formation of foot processes that are interconnected by the slit diaphragm, which is a critical component of the glomerular filtration barrier. In both experimental animals [23] and type II diabetes patients [24], reduction in the number of podocytes per glomerulus is associated with broadening of podocyte foot processes and is thought to contribute to the progression of diabetic nephropathy. Moreover, it is now widely recognized that proteinuria, specifically microalbuminuria, is one of the earliest clinically identifiable markers of diabetes-induced renal damage, and appearance of proteinuria indicates a compromised glomerular filtration barrier. A metaanalysis of glomerular transcriptomic characteristics of human and mouse samples of kidney diseases indicated a central role for the *EGFR* in both species [25].

We developed a podocyte-specific *EGFR* knockout mouse (EGFR*^podKO^*) by crossing *EGFR**^flox/flox^* mice [26] with Podocin Cre mice [27] (Figure 1). Effective cleavage of the *EGFR* gene was verified by PCR analysis of isolated glomerular genomic DNA, and effective deletion of *EGFR* protein expression in podocytes was confirmed by immunoblotting of isolated glomerular lysates [12]. In a model of type I diabetes induced by streptozotocin injection, the diabetic *EGFR**^podKO^* mice had significantly less albuminuria compared with the WT mice. Electron microscopy revealed more severe segmental podocyte foot process effacement in the WT mice, and there was significantly more podocytes loss in WT diabetic mice compared with the *EGFR**^podKO^* diabetic mice. To determine the mechanism of podocyte loss, we isolated glomeruli and analyzed the expression levels of Bcl2, an anti-apoptotic protein, and cleaved caspase3, a pro-apoptotic protein and found that the expression of Bcl2 was down-regulated, and the expression of cleaved caspase3 was up-regulated in WT diabetic glomeruli, and these alterations were significantly less in *EGFR**^podKO^* mice. The expression levels of TGFβ and fibronectin were up-regulated in the glomeruli isolated from WT diabetic mice, but there was marked inhibition of TGF-ß and fibronectin expression in *EGFR**^podKO^* diabetic mice.

To confirm the potential role of podocyte *EGFR* in the development of type II diabetic nephropathy, podocyte *EGFR* was selectively deleted in two mouse lines [28]. The first mouse line was *db/db* mice with selective podocyte *EGFR* deletion: nphs2-Cre; *EGFR*^f/f^; *db/db* (*egfr*^podKO^; *db/db*) mice and their corresponding controls, *egfr*^f/f^; *db/db* mice (*db/db*) mice. The second mouse line was an accelerated type II diabetic model with selective podocyte *EGFR* deletion: nphs2-Cre; *egfr*^f/f^; *nos*3^–/–^; *db/db* (*egfr*^podKO^; *nos*3^–/–^; *db/db*) mice and their corresponding controls, *egfr*^f/f^; *nos*3^–/–^; *db/db* (*nos3*^–/–^; *db/db*) mice. The effectiveness of *EGFR* deletion in podocytes was confirmed by determining genomic DNA), immunoblotting of glomerular lysates and immunofluorescent staining of *EGFR* expression. Most WT1 positive (Wilms Tumor Protein 1, a podocyte nuclear marker) cells were also *EGFR* positive in 20 weeks old *nos*3^–/–^; *db/db* mice. In contrast, most WT1 positive podocytes were devoid of *EGFR* staining or exhibited only faint *EGFR* staining in 20 weeks old *egfr*^podKO^; *nos*3^–/–^; *db/db* mice. Selective podocyte deletion reduced albuminuria in both models of type II DN and preserved podocyte number.

Podocytes have high basal levels of autophagy [29], and autophagy in podocytes has been implicated as important contributor to preserve podocyte structure and function [30], since genetic inhibition of autophagy in podocytes led to glomerulopathy [29,31,32]. Beclin-1 is an essential component of the autophagic machinery [33,34]. Glomeruli of *nos*3^–/–^; *db/db* mice had low levels of expression of beclin-1, but selective deletion of podocyte *EGFR* increased beclin-1 expression in podocytes as well as increasing LC3B puncta, a hallmark of autophagy. In addition, the autophagy substrate SQSTM1/p62, was decreased in glomeruli of *nos*3^–/–^; *db/db* mice, indicating increased autophagic activity. Rubicon binds to, and inactivates beclin-1, and rubicon expression was high and beclin-1 expression was low in glomeruli from *nos*3^–/–^; *db/db* mice while rubicon expression was lower and beclin-1 expression was higher in *egfr*^podKO^; *nos*3^–/–^; *db/db* mice. Knocking down rubicon in cultured podocytes using siRNA and treating with high glucose led to increased beclin-1 expression and autophagosome accumulation and decreased SQSTM1 expression. Immunofluorescent staining also confirmed that high glucose-induced rubicon upregulation and beclin-1 downregulation were abolished with inhibition of rubicon expression in podocytes.

mTOR activity increases in podocytes in diabetic mice, and correlates with increased ER stress and progressive glomerulosclerosis [35]. In addition to glomeruli, persistent mTOR activation has also been associated with apoptosis of renal tubule cells in diabetes [36]. Renal mTOR activation in poorly controlled diabetes may result from a combination of AKT inhibition of TSC2, hyperglycemia-induced AMP kinase inhibition and increased glucose uptake through glucose transporter 1 (GLUT1), in which the resulting increased glycolysis and activation of glyceraldehyde-3-phosphate dehydrogenase (GAPDH) can lead directly to Rheb activation of mTOR by reducing Rheb binding to GAPDH [37,38]. *EGFR* activation is a well-described mediator of mTOR activity through activation of the PI3K/AKT pathway [39,40]. In addition, *EGFR* activation inhibits renal gluconeogenesis and stimulates glycolysis in proximal tubules [41,42] and increases GLUT1 expression in mesangial cells [43]. Phosphorylation of mTOR and its partner raptor were markedly lower in erlotinib-treated than vehicle-treated STZ-eNOS^–/–^ kidney. In addition, erlotinib treatment led to decreases in phosphorylated p70 S6K and eIF-4B, downstream targets of mTOR signaling, and erlotinib treatment led to increased AMPK kinase activity. Immunolocalization indicated that phosphorylated AMPKα due to erlotinib treatment was increased in both glomeruli and in renal epithelial cells. Selective *EGFR* deletion in podocytes also led to decreases in renal mTORC activation, as indicated by decreased phosphorylation of the p-RPS6KB/p70S6k substrate RPS6 (ribosomal proteinS6) levels in *egfr*^podKO^; *nos3*^–/–^; *db/db* mice. Inhibition of *EGFR* activation or expression podocyte increased autophagy secondary to inhibition of mTORC1 and stimulation of AMP kinase [16,28].

## 6. The Role of *EGFR* in Tubulointerstitial Injury in Diabetic Nephropathy

In addition to the glomerulus, there is increasing evidence that the renal tubules and specifically the proximal tubules, are a target for diabetes and mediate the increased tubulointerstitial injury seen in diabetic nephropathy [44]. Diabetes-induced increased expression of *EGFR* and their ligands and subsequent *EGFR* activation have been previously reported both in in vivo models and in cultured renal proximal tubule epithelial cells [9,10]. Proximal tubule hypertrophy is an early response seen with the onset of diabetes, and in a streptozotocin model of type I diabetes, either erlotinib or selective proximal tubule deletion of *EGFR* expression reduced this early kidney hypertrophy.

The Hippo signaling pathway is a kinase cascade conserved from Drosophila to mammals that controls the balance of cell proliferation, cell differentiation and cell death to define organ size via regulating the phosphorylation and activation of YAP (Yes-associated protein) and/or TAZ (transcriptional co-activator with PDZ-binding motif), which serve as transcriptional co-activators for numerous target genes in the nucleus primarily by interacting with the TEAD family of transcription factors. Upon activation of the Hippo pathway in response to different extracellular cues, YAP and TAZ are phosphorylated at specific serine/threonine residues, which results in their inactivation by cytoplasmic sequestration and/or proteasome-mediated degradation, thereby inactivating downstream target genes expression, as indicated in Figure 1 [45]. YAP has moderate expression in normal adult kidney but very low expression in normal adult proximal tubule. TAZ, sharing 45% amino acid identity with YAP has higher expression in the normal adult kidney. YAP and TAZ play some redundant roles in the morula stage of mouse development and in controlling adult cardiac growth, but they also show differential functions in many aspects, as evidenced by embryonic lethality (at E8.5) with global deletion of YAP, whereas global mutation of TAZ caused early mortality in only a subset of homozygous mice, while surviving adult mice developed bilateral kidney cyst formation and a pulmonary emphysema-like phenotype.

YAP expression and activation (nuclear localization) increased in the proximal tubule in experimental models of both type I and type II diabetes and in kidney samples from patients with type II diabetes [13,46] while TAZ expression was either unchanged or actually decreased. The increased proximal tubule YAP expression and activation was inhibited in mice with proximal tubule *EGFR* deletion or by administration of the *EGFR* tyrosine kinase inhibitor, erlotinib. In cultured proximal tubule cells, high glucose increased nuclear association with TEAD and in the streptozotocin model increased renal expression of kidney expression of two known targets of YAP/TEAD gene transcription, CTGF and amphiregulin. Both CTGF and amphiregulin have been implicated in development of tubulointerstitial fibrosis [47,48]. Expression of both of these profibrotic factors was attenuated in *EGFR**^ptKO^* mice or by administration of erlotinib. *EGFR*-mediated nuclear translocation of YAP was dependent on RhoA/ROCK activation and subsequent activation of a PI3-kinase/AKT pathway.

To investigate the role of *EGFR* activation of YAP in mediation of proximal tubule injury in diabetic nephropathy. *Yap^PTiKO^* (inducible conditional proximal tubule-specific knockout of YAP) were utilized in the type I model of diabetes. Kidney expression of CTGF was decreased by proximal tubule selective deletion of YAP as well as the YAP/TEAD activation inhibitor verteporfin. Verteporfin decreased proteinuria, but there were no differences in proteinuria with selective proximal tubule Yap deletion. Either verteporfin or selective proximal tubule deletion of YAP significantly inhibited development of tubulointerstitial fibrosis. In vitro studies indicated that proximal tubule YAP activation resulted in secreted CTGF, which induced myofibroblast transformation in cultured fibroblasts.

## 7. Adverse Effects of *EGFR* Inhibitors

Although *EGFR* blocking antibodies and tyrosine kinase inhibitors are widely used in cancer therapy, there are serious concerns about whether they could ever be employed as therapeutic agents in kidney diseases, and specifically in diabetic nephropathy. The most common side effect seen in cancer patients treated with *EGFR* inhibitors is an acneiform rash, seen in between 50 and 100% of treated patients. In fact, the occurrence of the rash correlates with effectiveness of treatment. Although this side effect is deemed acceptable for the short-term treatment of life-threatening cancers, it would not be an acceptable side effect for what would be chronic continuous therapy in patients with diabetic nephropathy. In addition, magnesium wasting due to inhibition of magnesium reabsorption by the thick ascending limb is a rare complication of *EGFR* inhibition, especially with blocking *EGFR* antibodies

In summary, there is strong evidence for an important pathological role of persistent *EGFR* receptor activation in the development and progression of diabetic kidney disease. That inhibiting *EGFR* expression or activity can ameliorate progression of diabetic kidney injury indicates that the direct inhibition of *EGFR* activity and/or inhibition of signaling pathways activated by the receptor may be viable targets for prevention of progressive kidney injury resulting from diabetes.

## Figures and Tables

**Figure 1 cells-11-03416-f001:**
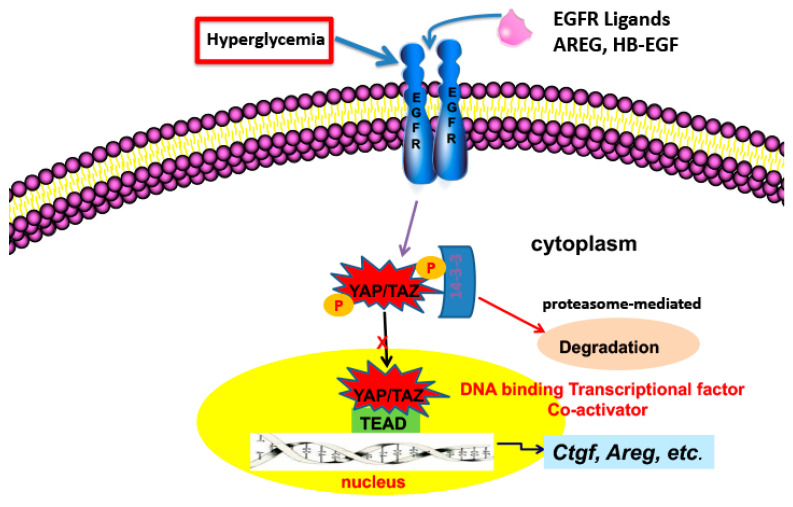
EGFR activation leads to increased YAP translocation to the nucleus, association with TEAD and transcription of genes involved in progressive kidney injury.

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
