# Peer review of "The Role of the Epidermal Growth Factor Receptor in Diabetic Kidney Disease"

_cells, 2022, doi:10.3390/cells11213416_

Round 1

Reviewer 1 Report

R.C. Harris reviewed the role of EGFR in diabetic nephropathy. The review is clearly written and the role of EGFR in DN in summarized based on the current knowledge.

I have 2 minor remarks:

Headings on the different chapters can make the review easier readable

The review could become even more interesting to the readership of the journal if the author discuss/summarize I bit more on how EGFR is dysregulated in diabetic patients, in particular the contribution of the hemodynamic factors, immunological factors and the metabolism on the EGFR signalling. The current review in particular summarize the importance of EGFR in DN. An extra (short) chapter which summarizes the factors which can upregulate EGFR in DN will strengthen the review.

Author Response

I appreciate the careful review and helpful comments.  In response , I have added headings for all of the topics covered, and I have included a new paragraph entitled “Mechanisms of Dysfunctional Expression and Activation of EGFR in Diabetic Kidney Disease”.

Reviewer 2 Report

The paper focuses on the importance of EGFR activation in the pathogenesis of DKD, and is relatively complete and informative. This review is well designed, however, I have a few comments for the authors.

1.The first appearance of English abbreviation should be labeled with the full name.

2.There has been some studies about renal adverse events in patients taking EGFR inhibitors. Brief summary of these contents can be provided.

3. Whether there are side effects for the inhibition of EGFR expression or activity should be discussed.

Author Response

I thank the reviewer for the helpful suggestions to improve the manuscript. In response, I have provided the full name of any abbreviations that I had not provided in the original version, and I have included a new paragraph entitled “Adverse Effects of EGFR Inhibitors.”

Round 2

Reviewer 2 Report

I have no further comments.